# Dickkopf-3: An Update on a Potential Regulator of the Tumor Microenvironment

**DOI:** 10.3390/cancers14235822

**Published:** 2022-11-25

**Authors:** Zainab Al Shareef, Mai Nidal Asad Ershaid, Rula Mudhafar, Sameh S. M. Soliman, Robert M. Kypta

**Affiliations:** 1College of Medicine, University of Sharjah, Sharjah P.O. Box 27272, United Arab Emirates; 2Research Institute for Medical and Health Sciences, University of Sharjah, Sharjah P.O. Box 27272, United Arab Emirates; 3College of Pharmacy, University of Sharjah, Sharjah P.O. Box 27272, United Arab Emirates; 4CIC BioGUNE, Basque Research and Technology Alliance, BRTA, Bizkaia Technology Park, 48160 Derio, Spain; 5Department of Surgery and Cancer, Imperial College London, London W12 0NN, UK

**Keywords:** Dickkopf-3 (Dkk-3) protein, DKK3 gene, tumor suppression, stroma, secreted glycoprotein

## Abstract

**Simple Summary:**

Dkk-3 is a secreted protein with context-dependent functions in several types of cancer. This review focuses on the roles of Dkk-3 in the tumor microenvironment and the effects of modulating Dkk-3 in different settings.

**Abstract:**

Dickkopf-3 (Dkk-3) is a member of the Dickkopf family protein of secreted Wingless-related integration site (Wnt) antagonists that appears to modulate regulators of the host microenvironment. In contrast to the clear anti-tumorigenic effects of Dkk-3-based gene therapies, the role of endogenous Dkk-3 in cancer is context-dependent, with elevated expression associated with tumor promotion and suppression in different settings. The receptors and effectors that mediate the diverse effects of Dkk-3 have not been characterized in detail, contributing to an ongoing mystery of its mechanism of action. This review compares the various functions of Dkk-3 in the tumor microenvironment, where Dkk-3 has been found to be expressed by subpopulations of fibroblasts, endothelial, and immune cells, in addition to epithelial cells. We also discuss how the activation or inhibition of Dkk-3, depending on tumor type and context, might be used to treat different types of cancers.

## 1. Introduction

The expression of the gene encoding Dickkopf-3 (*DKK3*; chromosome 11p15 in humans) is reduced in many tumor types as well as in many cancer cell lines [1] and in immortalized cells, giving rise to the alternative name *REIC* (Reduced Expression in Immortalized Cells) [1]. The loss of *DKK3* gene expression in cancer is frequently a consequence of gene promoter methylation, which varies between 14% in non-small cell lung cancer and 78% in breast cancer [2]. In contrast, in some other cancers such as oral squamous cell carcinoma (OSCC) and ganglioneuroma, the *DKK3* promoter is not hypermethylated and these cancers express high levels of Dkk-3 [3,4]. Reversal of *DKK3* gene promoter methylation by demethylating reagents such as decitabine and zebularine or using a CRISPR-based approach results in increased Dkk-3 protein levels and can reduce tumor cell proliferation and migration [5,6,7,8]. The inhibitory effects of Dkk-3 on tumor cell proliferation have been exploited in adenoviral therapies (Ad-REIC), where promising results have been observed in prostate cancer patients [9]. In addition to the pro-apoptotic effect of the high expression of Dkk-3 as a result of endoplasmic reticulum (ER) stress, the therapeutic benefits of Dkk-3 have also been proposed to involve a distant bystander effect through the stimulation of the immune system [10].

Dkk-3 contributes to several biological processes that implicate a role in cell and tissue differentiation. These include the maintenance of cartilage [11], endothelial regeneration [12], and the induction of myofibroblasts from fibroblasts (Figure 1) [13]. In addition, Dkk-3 has been reported to be involved in the differentiation of induced pluripotent and embryonic stem cells [12]. Although Dkk-3 inhibits cancer cell proliferation [14,15,16], its function in stromal cells, particularly in a cancer setting, is less clear. One aim of this review is to highlight the potential role of Dkk-3, and in particular, stromal Dkk-3, as a component of the tumor microenvironment. Some of the roles of Dkk-3 in cancer are summarized in Table 1.

## 2. Mechanistic Contribution of Dkk-3 to Cancer Suppression

The *DKK3* gene is a member of a family comprising DKK1-4 and DKKL1/Soggy [21]. DKK1-4 encodes two cysteine-rich domains (CRD1 and CRD2, also known as Cys1/N and Cys2/colipase fold). A stable ‘core region’ containing CRD1 and CRD2 has been reported to promote anticancer immunity [22]. While DKKL1 shares sequence homology with DKK3, it does not encode CRDs [21]. Dkk-1/2/4 directly inhibits Wnt/β-catenin signaling by binding to the Wnt co-receptors LRP5/6 (low-density lipoprotein receptor-related proteins 5 and 6) [23,24]. This is not the case for Dkk-3, which lacks key amino acids in CRD2 involved in binding. Dkk-1/2/4 can also associate simultaneously with LRP5/6 and the co-receptor Kremen (Krm1/2) [23]. To date, only cytoskeleton-associated protein 4 (CKAP4) has been demonstrated to bind Dkk-1/2/4 and Dkk-3, an interaction mediated by CRD1 [25,26]. Dkk-3 CRD2 appears to have biological activity [27] (see below), but receptors that directly bind to this domain are yet to be identified.

Although the role of Dkk-3 in Wnt/β-catenin signaling is not the same as for Dkk-1/2/4 as it is unable directly to bind to the Wnt co-receptor LRP6 or its partner Kremen (Krm) [21,28], there is evidence for Dkk-3 and Wnt signaling cross-talk. Notably, reducing the expression of DKK3 increases Wnt signaling activity in breast cancer cells and potentiates the effect of β-catenin in prostate cancer cells [29,30]. The mechanism by which Dkk-3 inhibits Wnt signaling may be indirect, through β-transducin repeats-containing protein (β-TrCP), which normally promotes β-catenin degradation. This interaction was first observed in a yeast-2-hybrid screen [31] and its significance was initially unclear, as Dkk-3 is secreted [32] and β-TrCP is found in the cytosol and nucleus [33]. However, immunolocalization and subcellular fractionation studies revealed that Dkk-3 and β-TrCP co-localize in the cytosol and their interaction attenuates Wnt signaling by blocking β-catenin translocation to the nucleus [29]. Besides β-TrCP, Dkk-3 has been reported to bind Krm intracellularly, possibly in the endoplasmic reticulum or the Golgi [34]. Additionally, an intracellular splice-form of mouse Dkk-3 has been identified and demonstrated to inhibit Wnt/β-catenin signaling by directly binding to β-TrCP [35]. Recombinant human Dkk-3 was also shown to bind to β-TrCP and inhibit Wnt/β-catenin signaling in iPS-derived embryoid bodies [35]. In prostate, lung, and breast cancers, Dkk-3 mediates cancer cell apoptosis via the activation of JNK phosphorylation as a subsequence of the cleavage of caspases 9 and 3 [36,37,38]. Therefore, Dkk-3 possesses cell specific activation of downstream signaling.

Tumor progression and nuclear β-catenin expression have been linked to a reduction in DKK3 gene expression in lung tumors, as a result of DKK3 promoter methylation. Ectopic expression of DKK3 in lung cancer cells reduces T-cell factor (TCF)-4 and β-catenin signaling activity and induces apoptosis [39]. Ectopic DKK3 expression also induces apoptosis in prostate cancer cells. This has been attributed to a stress response induced by the N-terminal 78 amino acid residues of Dkk-3 [40,41]. A recent study by Zhao et al. found that DKK3 overexpression and recombinant Dkk-3 protein inhibited proliferation by triggering G2 cell cycle arrest and increased apoptosis by reducing mitochondrial membrane potential in colorectal cancer cells [42]. The latter is consistent with an earlier study showing a positive relationship between DKK3 overexpression, caspase-3 activity (the major effector of cytochrome c), and the mitochondrial translocation of BAX [38]. In order to test the endogenous tumor suppressor activity of mouse Dkk3, Zhao et al. crossed conditional Dkk3 knockout and villin-cre mice, observing a chemically induced colorectal adenoma in one of the five Dkk3 mutant mice examined [42]. At present, there are few examples of partners for the secreted form of Dkk-3. However, a recent study found that Dkk-3 CRD1 interacts with the ECM protein, transforming the growth factor beta-induced (TGFBI) [43], a secreted protein that has pro-invasive features and has been classified as a tumor-promoting factor [13,44]. A link between Dkk-3 and TGFBI was previously observed in prostate cancer [13] (see below). The nature of the interaction between Dkk-3 and TGFBI and their potential signaling crosstalk is complex and possibly tissue-specific, with reduced expression of Dkk-3 in prostate cancer leading to increased expression and secretion of TGFBI [13] and the binding of TGFBI inhibiting the effects of Dkk-3 on integrin-dependent adhesion in hepatocellular carcinoma [43].

## 3. Dkk-3 Receptors

Cell proliferation is mediated by CKAP4, which as noted above, is a receptor for DKK family proteins. CKAP4 may be involved in a variety of processes in addition to proliferation, migration, and invasion [25,26]. CKAP4 has anti-proliferative effects in bladder carcinoma cells, where it stimulates the inhibitory effects of anti-proliferative factor (APF) and upregulates TP53 expression, activates AKT/GSK3β/β-catenin-dependent pathways, and downregulates MMP2 expression [45]. Since DKK3 is highly expressed in bladder carcinoma cells and acts as a ligand for CKAP4, it might function as a prognostic biomarker in this setting [26]. About half of the tumor lesions of patients with esophageal squamous cell carcinoma (ESCC) express Dkk-3. Anti-CKAP4 antibodies that inhibit the binding of Dkk-3 to CKAP4 reduce tumor formation induced by ESCC cells in xenograft assays. Anti-CKAP4 antibodies also inhibit the growth of esophageal organoids. These exciting findings suggest a Dkk-3-CKAP4 axis that might serve as a novel molecular target for ESCC [26,46].

The binding of Dkk-3 to CKAP4 is mediated by Dkk-3 CRD1. However, CRD2 also appears to be important for Dkk-3 function. For example, it is required for the Dkk-3 inhibition of TGF-β-dependent migration and invasion in PC3 metastatic prostate cancer cells [27], suggesting that this domain might be useful to block the tumor-promoting effects of TGF-β signaling in advanced prostate cancer [27]. The role of Dkk-3 in the remodeling and regeneration of damaged tissues could involve other receptors, for example, Dkk-3 binding to the chemokine receptor CXCR7 has been reported to activate vascular progenitor cell migration and induce tissue vessel regeneration in vitro and in blood vessels [47].

## 4. Dkk-3 as a Regulator of Tumor Stromal Cells

### 4.1. The Role of Dkk-3 in the Regulation of Cancer Fibroblasts and Stellate Cells

The fundamental role of the stroma in regulating cancer progression has been reported in several settings [48,49]. The ability of stromal cells to either accelerate or attenuate cancer cell proliferation, invasion, metastasis, and even affect response to treatment are depicted in Figure 2 [50,51]. Tumor microenvironment cells may be fibroblasts, endothelial cells, pericytes, or bone marrow-derived cells such as mesenchymal stem cells (MSC), neutrophils, macrophages, and mast cells [51,52]. Signaling crosstalk among cancer cells, stromal cells, and epithelial cells is mediated by chemokines, cytokines [53], and growth factors and proteases that remodel the extracellular matrix [52,53]. In general, fibroblasts are the predominant cell type in the stromal compartment of most solid tumors [53]. Cancer-associated fibroblasts (CAFs), also called cancer-reactive stromal cells, become activated to proliferate and increase the production of ECM, acquiring features of smooth muscle (SM) cells, similar to myofibroblasts [53]. Recently, Dkk-3 was reported to modulate the response to TGF-β, which promotes the differentiation of fibroblasts into CAFs in prostate cancer [13]. This link may reflect the interaction between cancer/stromal cell compartments and epithelial–mesenchymal transition (EMT) progression, since TGF-β can be shifted from being anti-tumorigenic to potently pro-tumorigenic [54,55].

An association between Dkk-3 and tumor stroma has been reported in ER-negative breast cancer, ovarian cancer, and colon cancer cells [17], where *DKK3* gene expression is upregulated in the tumor stroma, correlating with a more aggressive tumor type [17]. Heat shock factor-1 (HSF-1) was found to regulate *DKK3* gene expression in stromal cells as part of the response to stress including oxidative stress, nutrient deprivation, and protein misfolding. This study further reported that stromal Dkk-3 enhances pro-tumorigenic signaling by activating YAP/TAZ (Figure 3A) [17].

In benign prostatic hyperplasia, Dkk-3 is more highly expressed in the prostate stroma, particularly in myofibroblasts, compared to the epithelium [13,56]. This alteration may be a response to the loss of Dkk-3 in the epithelium, which normally opposes the potential tumorigenic impact of TGF-β by reducing Smad3 phosphorylation in myofibroblasts (Figure 3B); thus, stromal Dkk-3 may contribute to supporting a normal acinar architecture, thereby reducing proliferation and limiting prostate cancer cell invasion [13]. Notably, these changes in the expression of Dkk-3 in the two compartments, epithelium, and stroma, are inversely correlated with changes in the expression of TGFBI [13]. As noted earlier, it has also been proposed that Dkk-3 positively regulates Wnt/β-catenin signaling through an interaction with Krm (Figure 3C) [34]. Another study reported the therapeutic effect of Ad-REIC, a DKK3-expressing adenovirus being developed for gene therapy, through the inhibition of CD147 (cluster of differentiation 147) [57]. CD147 is a surface glycoprotein upregulated in many solid tumors including prostate cancer. Cell surface expression of CD147 in cancer promotes adjacent fibroblasts and cancer cells to secrete matrix metalloproteinases (MMPs), which are inducers of cancer cell invasion and metastasis [57]. At the same time, CD147 induces VEGF and hyaluronan, thereby stimulating angiogenesis, drug resistance, and anchorage-independent growth. Ad-REIC treatment significantly reduces CD147 levels in prostate cancer cell lines [57]. In adult human osteoarthritis, an inflammatory disease, increased expression of DKK3 may protect against cartilage degradation and aberrant signaling associated with this disease [11]. Dkk-3 reduces metalloproteinases by attenuating NFκB signaling, thus preventing the breakdown of cartilage, which leads to irreversible degradation of ECM, especially type 2 collagen [58]. The differential expression of DKK3 in different cancers compared to normal cells is shown in Figure 4.

Dkk-3 has been also described as a tumor-promoting factor, especially in cancers associated with the upregulation of *DKK3* expression such as in head and neck and pancreatic cancers (Figure 4) [59,60]. For example, a study conducted on human pancreatic duct adenocarcinoma (PDAC) and normal pancreas reported that Dkk-3 is mainly expressed in the stroma, where it is secreted by pancreatic stellate cells (PSC) [19]. Dkk-3 was found to enhance PDAC metastasis and interfere with chemotherapy resistance, while a combination of a Dkk-3 function blocking antibody, JM6-6-1, and immune checkpoint inhibitors was found to limit tumor growth and survival [19]. It is a common concept that PSCs play a role in promoting PDAC proliferation and survival as well as reducing PDAC cell responses to therapy [19]. Thus, silencing of DKK3 in human PSCs was shown to inhibit PDAC cell proliferation and migration and blocking Dkk-3 might contribute to better responses to chemotherapy [19].

### 4.2. The Role of Dkk-3 in the Regulation of Cancer Angiogenesis

Dkk-3 may play a role in the remodeling of stroma through its involvement in angiogenesis [12,61]. Angiogenesis is fundamental for cancer invasion and metastasis [62]. Dkk-3 has been reported to affect the expression levels of angiopoietin 1 (ANGPT1), angiopoietin 2 (ANGPT2), and vascular endothelial growth factors (VEGF) [18,47,61]. During tumor progression, the ANGPT switch is considered to be rate-limiting in favor of a higher ANGPT2/ANGPT1 expression ratio [63]. In benign prostatic hyperplasia (BPH) and prostate cancer, Dkk-3 increases ANGPT2 expression, a destabilizing factor that leads to the formation of vessel sprouting in the tumor microenvironment [18]. Dkk-3 also supports human umbilical vein endothelial cell (HUVEC)-related tube formation through increasing VEGF stimulation by activating both ALK1 and TGF-β/Smad signaling [64]. Busceti et al. (2018) used cultured endothelial cells to show that Dkk-3 induces VEGF expression to exert a protective effect against ischemic neuronal injury [65]. Moreover, Dkk-3 has been reported to enhance new vessel growth in non-MYC-driven medulloblastoma [20]. Interestingly, Dkk-3 is also an inducer of cavernous vascular endothelial cells, pericytes, and endothelial junctional proteins such as claudin-5 and ZO-1 in diabetic mice with erectile dysfunction (ED), increasing cavernous endothelial cell expression of angiogenesis factors including ANGPT1, VEGF, and bFGF. Thus, Dkk-3 might be useful as a therapy for diabetic patients with ED [66]. Dkk-3 also promotes the migration and recruitment of Sca1^+^ vascular stem/progenitor cells from the murine aortic adventitia. As already noted, Dkk-3 mediates vascular progenitor cell migration by binding to the chemokine receptor CXCR7 [47]. Dkk-3/CXCR7 engagement in vascular progenitor cells appears to lead to the activation of ERK1/2, phosphatidylinositol 3-kinase/AKT, and the small GTPases Rac1 and RhoA [47].

Dkk-3 not only influences angiogenesis but may also be useful in regenerative medicine, where it has been found to rejuvenate ischemic tissue. This study reported that the expression of DKK3 could create functional endothelial cells from fibroblasts by activating a “mesenchymal to endothelial differentiation” VEGF/miR-125a-5p/Stat3 signal [12]. In addition, Dkk-3 can promote endothelial cell regeneration, which contributes to protection against atherosclerosis [12]. In contrast, in renal fibrosis, secreted Dkk-3 is described to contribute to trans-differentiation of endothelial cells to myofibroblasts, termed “endothelial to mesenchymal differentiation” in vivo [67].

### 4.3. The Role of DKK3 in the Regulation of Cancer Immune Responses

Dkk-3 is not only involved in the activation of fibroblasts to myofibroblasts and angiogenesis but also modulates the inflammatory and immune responses of inflammatory cells including macrophages, mast cells, eosinophils, and neutrophils, which are part of the stromal compartment [68]. During inflammation, inflammatory cells are stimulated to release cytokines and chemokines that stimulate the initiation of the immune response. Persistent or chronic inflammation may incite carcinogenesis by causing DNA damage, cell proliferation, and angiogenesis [68]. Non-alcoholic fatty liver is a chronic inflammatory liver disease that is exacerbated in liver-specific Dkk3 knockout mice and ameliorated in liver-specific Dkk3-overexpressing transgenic mice, in a mechanism that requires the MAP kinase Ask1 [69]. Studies in mouse models of chronic renal inflammatory disease, in contrast, found that Dkk3 induces cytokines that lead to fibrosis [70]: Dkk3 gene knockout or treatment with a Dkk3 function-blocking antibody was accompanied by increased accumulation of IFNγ-producing Th1 and Tregs at the expense of Th2 cells, which are profibrotic [71,72]. A third example from studies in mice is in the pancreas, where Dkk3 levels are increased in caerulein-induced acute and chronic pancreatitis. In this context, Dkk3 knockout increased recovery rates by limiting canonical Wnt signaling and increasing Hedgehog (Hh) signaling during pancreatic regeneration. These effects have been proposed to involve a reduction in the expression of the transcriptional repressor Gli3 [73]. The same study reported a similar scenario in the context of liver regeneration, leading the authors to propose that Dkk3 functions as a roadblock in liver and pancreas repair/regeneration upon injury [73]. Dkk-3 activation of non-canonical Wnt/β-catenin-independent signaling involving c-JUN N-terminal kinase (JNK) increases the secretion of inflammatory factors, such as TNF-α and IL-1β, in the brain, suggesting a neuroprotective mechanism [74] Furthermore, Dkk-3 is a factor required for smooth muscle cell differentiation [74]; the absence of Dkk-3 may therefore contribute to accelerating blood vessel inflammation and subsequent atherosclerotic plaque formation. At the same time, overexpressed Dkk-3 ameliorates myocardial infarction and inhibits the associated inflammatory process through the regulation of ASK1–JNK/p38 signaling [74].

The expression of Dkk-3 in mesenchymal stem cells also limits CD8^+^ and CD4^+^ T cell-mediated responses to suppress the process of inflammation [75]. Mesenchymal stem cells are known to suppress the growth of tumors by inhibiting T cell proliferation, the induction of T regulatory cells, and facilitating the generation of immunosuppressive M2-type macrophages [75]. Treating mice with intracerebral hemorrhage with recombinant Dkk-3 results in a significant reduction in the release of TNF-α, cleaved caspase-1, and IL-1β. Dkk-3 thus appears to reduce JNK/AP-1-mediated inflammation, which may improve the neurological outcomes [76].

Human fibroblasts infected with Ad-REIC showed increased IL-7 production, which indirectly inhibits cancer tumorigenesis [77,78]. Furthermore, the expression of Dkk-3 ameliorates graft transplantation through the regulation of T cell-mediated rejection [79]. In addition to the induction of IL-7 secretion, Ad-REIC facilitates an anti-tumorigenic microenvironment by promoting tumor-associated antigen-specific CD8^+^ cytotoxic T lymphocytes (CTLs) [9]. Injection of Ad-REIC leads to the presentation of proteins derived from apoptotic cancer cells to dendritic cells (DCs), thereby inducing the conversion to cytotoxic T cells and CD8^+^ CTLs [9]. On the other hand, deletion and/or neutralization of Dkk3 in TCR/MHC class-I double transgenic mice increases local CD8^+^ T cell infiltration and enhances MHC class-I mismatched anti-tumor and skin graft rejection [79]. Thus, organs that are classified as tissues with limited regenerative capacity such as the nervous system, eye, uterus, and placenta show the highest Dkk3 expression, possibly creating an immunosuppressive microenvironment and protecting against autoimmunity [80]. In addition, a Dkk3^−/−^ experimental autoimmune encephalitis (EAE) CNS mouse model showed an increase in the total number of infiltrating T cells, particularly the IFNγ-producing CD4^+^ and CD8^+^ T cells, leading to disease exacerbation. In this EAE and Dkk3 deficiency model, the hypothesis is that the action of Dkk3 is local and does not include secondary lymphoid organs [79]. In human glioblastoma, a machine-learning approach found an inverse correlation between DKK3 gene expression and anti-tumoral immunity, particularly in CD8^+^ and CD4^+^ T cells [81]. These observations merit further studies of mouse and human tumor models to clarify the role of Dkk-3 as a tissue-derived immune modulator and potential immunotherapy. Of potential relevance, serum autoantibodies to cancer/testis antigens (CTAs), which are used as biomarkers for the anti-tumor immune response, were recently measured in Ad-REIC-treated CRPC patients and it was concluded that monitoring CTAs in this setting can contribute to clinical management [82].

While most mouse studies have highlighted potential effects on T cells, Dkk3 is also a player in B cell-mediated autoimmune disease. B cells are divided into B1 and B2 subsets, with distinct functions, time of maturation, and anatomical location [83,84]. B1 cells secrete the majority of natural immunoglobulins (IGs) in the innate immune response, while B2 cells are responsible for adaptive immune responses [84]. Dkk3-deficient mice show suppression of B2 cell maturation and reduced proliferation and self-maintenance of peripheral B1 cells [83].

### 4.4. The Role of DKK3 in Stem Cell Differentiation

The role of Dkk-3 in orchestrating the quiescence and/or the differentiation of stem/progenitor cells has been investigated in prostate, breast, pancreatic, liver, and oral submucosal diseases [73,85,86,87]. Over the last decade, it has been of interest to identify the genes of somatic cells involved in re-programming to improve organogenesis for tissue engineering [88]. In this context, Dkk-3 has been highlighted as a molecule with overlapping regulatory effects on the Wnt and Hh signaling pathways during tissue regeneration and somatic reprogramming [73]. As noted above, studies in mouse models found that loss of Dkk3 ameliorates liver and pancreatic injury in acute and chronic disease via increased expansion and differentiation of the liver progenitor cell pool [73]. Additionally, Dkk3 induces the differentiation of mouse embryonic stem cells to smooth muscle cells [89]. Consistent with a role for Dkk-3 in human progenitor cell differentiation, silencing of DKK3 in benign prostate epithelial (RWPE-1) and stromal (WPMY-1) cell lines revealed reductions in the expression of the stem cell marker SOX2 [13]. In prostate stromal cells, DKK3 silencing also reduced the expression of the stem/progenitor cell marker *s-SHIP* [85] as well as other putative stem/progenitor cell markers [13]. Importantly, Dkk3 was identified as one of the top hits in a shRNA screen to identify modulators of mouse embryonic fibroblast reprogramming to generate induced pluripotent stem cells (iPSCs) [73], providing further evidence for the contribution of fibroblast DKK3 in the regulation of cell fate [17,19]. Mesenchymal stem cells (MSCs), which are thought to be progenitors of myofibroblasts in the cancer-associated stroma [90], secrete soluble factors that reduce undesirable immune reactions such as hypersensitivity, autoimmune disease, and graft rejection [91]. Importantly, MSCs secrete Dkk3 [75], and tumors containing Dkk3^−/−^ MSCs show increased CD8^+^ T cell invasion and reduced M2-type macrophage infiltration, suggesting MSC-derived Dkk3 maintains the immune-suppressive capacity of the tumor microenvironment [75]. Moreover, MSCs exposed to radiotherapy showed increased expression and secretion of Dkk-3, and MSCs were shown to enhance the effects of radiotherapy on tumor growth in a mouse melanoma xenograft model [92].

## 5. Diagnostic and Therapeutic Potential of DKK3 in Cancer

A wealth of data from in vitro and xenograft cell models in multiple cancer types have correlated changes in DKK3 expression with disease progression. For example, the assessment of DKK3 expression together with a CT scan image improves metastasis prediction in gastric cancer patients [93]; DKK3 gene promoter hypermethylation, which is associated with reduced gene expression, correlates with disease recurrence in esophageal cell carcinoma [94] and with disease recurrence and metastasis in breast [95] and gastric cancers [6]. In contrast, low DKK3 mRNA expression in esophageal adenocarcinoma patient samples is associated with an unfavorable treatment response to neoadjuvant chemo-radiotherapy (CRT) [96]. The prognostic value of DKK3 expression has also been reported in ovarian [97], renal [98], and thyroid [99] cancers and in blood cancers including acute myeloid leukemia [100]. Recently, a digital droplet methylation specific polymerase chain reaction (ddMSP) assay was developed to detect DKK3 promoter methylation in the serum from malignant mesothelioma patients [101]. The authors detected 5–500 copies of methylated *DKK3* in 4 mL of serum from eight out of 21 patients and there was a trend for increased methylation in patients with more advanced disease [101]. However, as noted earlier, the correlation of DKK3 expression and disease progression is dependent on tumor type, and most examples from genetic mouse models highlight the pro-tumorigenic effects of Dkk-3 secreted by non-tumor cells, in particular, stromal and immune cells. This may underscore differences between man and mouse, or, more likely, the importance of taking a holistic approach when considering how Dkk-3 might be targeted for therapy. Regardless of the case, the ongoing development of therapies that either activate (DKK3-expressing adenoviruses) or inhibit (function-blocking anti-Dkk-3 antibodies) Dkk-3 make for a deeper understanding of Dkk-3 function in the clinical context ever more pertinent. Dkk-3 levels in blood and urine have been observed in non-cancer settings, in particular, Dkk-3 plasma levels are associated with cardiovascular risk factors including age, male sex, body mass index, and glucose levels [102]. Serum Dkk-3 is also a potential biomarker for tissue fibrosis for lymphatic nephritis associated with systemic lupus erythematous disease (SLE) [103], and increased urinary Dkk-3 can identify patients with chronic kidney disease who are at risk for loss of kidney function [104]. These diseases may confound attempts to use the detection of Dkk-3 to monitor cancer progression [105].

### 5.1. Epigenetic Reactivation of Dkk-3 Expression as a Cancer Therapy

DKK3 expression is downregulated in several types of human cancers [106]. This is frequently associated with the hypermethylation of the promoters of DKK3 and other tumor suppressor genes [16]. DNA hypomethylating reagents such as 5-Aza-2′-deoxycytidine (decitabine) have been proposed as potential therapeutics [16]. Indeed, decitabine induces DKK3 expression in prostate cancer [8] and AML cells [107]. However, there are obvious clinical challenges including non-selective cytotoxicity, poor bioavailability, and transient activity as DNA methylation levels return to normal once the drug is withdrawn [106].

The Clustered Regularly Interspaced Short Palindromic Repeats (CRISPR) associated protein 9 (Cas9) system is a valuable tool in the discovery and development of drugs against essential targets and can be exploited to modulate the expression of endogenous genes [108,109]. Kardooni et al. used CRISPR to reactivate DKK3 gene expression in prostate cancer cells, finding that this inhibited TGF-β-dependent signaling and tumor cell migration [8]. A more advanced CRISPR system employing an adenoviral vector has been generated that facilitates light-inducible induction of DKK3 gene expression in vivo, providing a potential selective and safe form of gene therapy. The effectiveness of this system was demonstrated in a xenograft tumor model [110].

### 5.2. Ad-REIC Gene Therapy

The initial report describing the Reduced Expression in Immortalized Cells (REIC) gene from Okayama University, Japan in 2000 came from a study of immortalized human fibroblast cells [1]. Thereafter, REIC was found to be identical to human DKK3 and found not to be expressed in the majority of cancer cell lines. Since then, *DKK3* has been described to be a tumor suppressor gene [111,112,113,114,115,116,117], although this is clearly not generally the case in some contexts. An adenovirus expressing Dkk-3 (Ad-REIC) strongly inhibits prostate tumor xenograft growth and metastasis in immunocompromised mice, increasing their survival [118]. Based on successful in vitro and in vivo studies, Ad-REIC was taken to clinical trials. Ad-REIC was classified as a safe medication with no overt side effects [9]. Furthermore, Ad-REIC exhibited anti-tumor activity at distant tumor sites via effects on IL-7, which promotes natural killer (NK) cell infiltration [10,78]. The clinical trial is ongoing (NCT01931046). There was an interesting case study of a 63-year-old patient diagnosed with a Gleason 9 (4 + 5) tumor with multiple lymph node (LN) metastases and extremely high PSA levels (483 ng/mL). Upon Ad-REIC treatment, there was a decline in PSA levels and a resolution of metastasis after additional Ad-REIC injections. No side effects were noticed, and the trial was conducted for two years [9]. Additionally, Ad-REIC enhances the treatment response in glioma [119] and there is an ongoing Phase I/IIa clinical trial in glioma patients [120,121]. Likewise, liver phase I/Ib clinical trial [122], malignant mesothelioma phase II clinical trials (ID: NCT04013334), and glioma [121] are underway. However, this gene therapy approach may be limited by the vectors used. The adenoviral vector is for short-term expression, and it can trigger strong host immune reactions. Thus, the development of a safe and longer-term Dkk-3 expression and delivery system is still under investigation [123]. Furthermore, understanding the molecular mechanism of action of Dkk-3 may aid in identifying those patients who will respond better to a Dkk-3-based therapy.

## 6. Conclusions

Dkk-3 appears to play a beneficial role for patients in many, but not all cancers, and how it functions remains poorly understood. The answer may lie in the identity and expression patterns of its receptors. Although cell-surface receptors that mediate the beneficial effects of Dkk-3 have yet to be identified, some evidence points to β-TrCP as an effector for the inhibitory effects of intracellular Dkk-3 on β-catenin signaling [35]. Dkk-3 is pro-tumorigenic in settings where its receptors such as CKAP4 in ESCC [47] and CXCR7 in blood vessels are involved [47]. Through its varied effects on multiple cell types in the tumor microenvironment including fibroblasts, immune cells, and vascular cells, Dkk-3 appears to have the ability to enhance or reduce cancer progression by altering tumor cell responses to signals mediated by Wnts (β-catenin-dependent and -independent via JNK), TGF-β, and Hh. In addition, Dkk-3 can influence stem/progenitor cell differentiation during tissue regeneration and somatic reprogramming. Taken together, translational studies that assess and modulate Dkk-3 expression and activity may lead to new avenues for the prognosis, prevention, and treatment of cancer and non-cancer diseases. However, if they are to be successful in the clinic, approaches currently being developed that harness the anti-tumorigenic properties of Dkk-3 will need to consider the contexts where Dkk-3 appears to be pro-tumorigenic. Furthermore, understanding the underlying mechanisms, specifically the tissue-specific receptors and the dominant stimulatory pathways in each setting will contribute to the discovery of novel Dkk-3-based therapies.

## Figures and Tables

**Figure 1 cancers-14-05822-f001:**
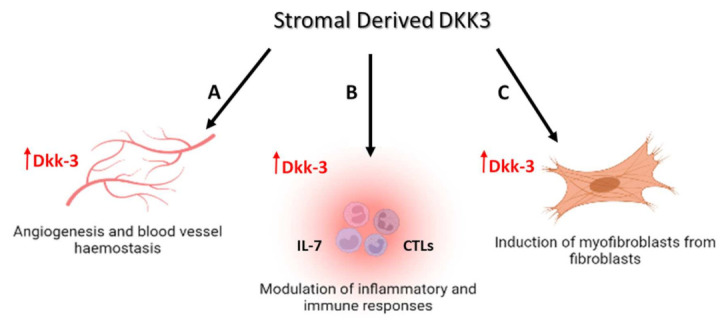
Participation of DKK3 in multiple cell and tissue settings. (**A**) Upregulated stromal Dkk-3 in blood vessels destabilizes angiopoietin 1 (ANGPT1) and promotes microvessel formation (according to Zenzmaier et al., 2013). (**B**) Dkk-3 promotes IL-7 secretion and increases CD8+ cytotoxic T lymphocyte (CTLs) infiltration, and hence inhibits cancer tumorigenesis (according to Kumon et al., 2016 and Watanabe et al., 2014). (**C**) Upregulated stromal Dkk-3 induces the trans-differentiation of fibroblasts to myofibroblasts and increases fibroblast proliferation (according to Zenzmaier et al., 2013).

**Figure 2 cancers-14-05822-f002:**
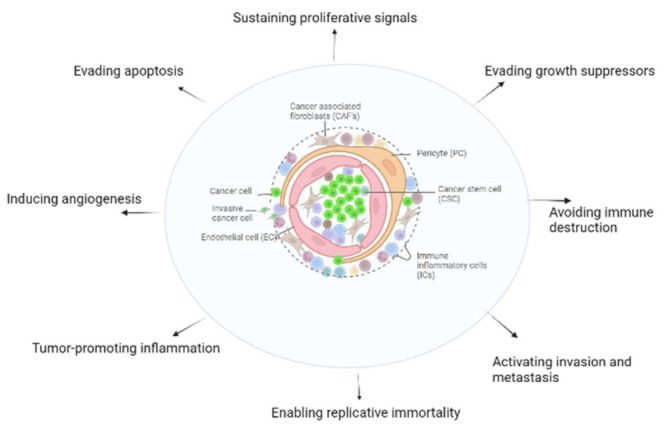
Roles of cancer stromal cells in the hallmarks of cancer.

**Figure 3 cancers-14-05822-f003:**
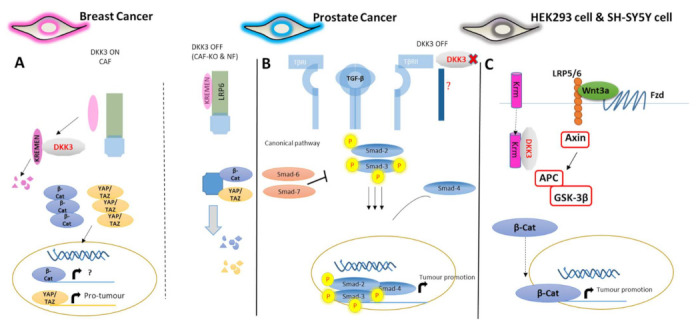
Potential mechanisms of action of Dkk-3 in different cancers. **(A)** Intracellular Dkk-3 in breast cancer-associated fibroblasts (CAFs) destabilizes Kremen, leading to the potentiation of YAP/TAZ and β-catenin signaling, adapted from Ferrari et al. (2019). (**B)** Dkk-3 secreted by WPMY-1 prostate stromal cells reduces TGF-β-dependent invasion of prostate cancer cells. The Dkk-3 receptor remains unidentified. (**C**) Proposed mechanism of action of Dkk-3 based on results from human embryonic kidney (HEK293) and neuroblastoma (SH-SY5Y) cell line studies. Dkk-3 potentiates Wnt signaling by reducing Krm plasma membrane levels, Dkk-3 interacts with Krm but not LRP5/6 intracellularly (e.g., in the Golgi or ER); adapted from Nakamura et al. [42].

**Figure 4 cancers-14-05822-f004:**
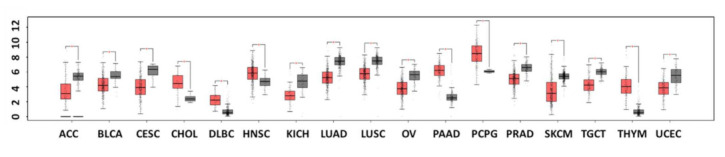
Differential DKK3 expression in cancer versus normal cells using RNA-Seq analysis of 9736 tumors and 8587 normal samples from TCGA and GTEx. ACC; Adrenocortical carcinoma (num(T) = 77; num(N) = 128), BLCA; Bladder Urothelial Carcinoma (num(T) = 404; num(N) = 28), CESC; Cervical squamous cell carcinoma and endocervical (num(T) = 306; num(N) = 13), CHOL; Cholangiocarcinoma (num(T) = 36; num(N) = 9), DLBC; (num(T) = 47; num(N) = 337), HNSC; Head and Neck squamous cell carcinoma (num(T) = 519; num(N) = 44), KICH; Kidney Chromophobe (num(T) = 66; num(N) = 53), LUAD; Lung adenocarcinoma (num(T) = 483; num(N) = 347), LUSC; Lung squamous cell carcinoma (num(T) = 486; num(N) = 338), OV; Ovarian serous cystadenocarcinoma (num(T) = 426; num(N) = 88), PAAD; Pheochromocytoma and Paraganglioma Thymoma (num(T) = 179; num(N) = 171), PCPG; (num(T) = 182; num(N) = 3), PRAD; Prostate adenocarcinoma (num(T) = 492; num(N) = 152), SKCM; Skin Cutaneous Melanoma (num(T) = 461; num(N) = 558), TGCT; Testicular Germ Cell Tumors (num(T) = 137; num(N) = 165), THYM; Thyroid carcinoma (num(T) = 118; num(N) = 339), UCEC; Uterine Corpus Endometrial Carcinoma (num(T) = 174; num(N) = 91). The data were prepared using http://gepia.cancer-pku.cn (accessed on 10 September 2022).

**Table 1 cancers-14-05822-t001:** The role of DKK3 in the modulation of the cancer microenvironment and its therapeutic aspects.

Tumor Origin	DKK3 in Tumor Stroma	Therapeutic Potential	Reference
**Breast**	➢ Significantly upregulated with cancer aggressiveness, particularly in ER-negative breast cancer. ➢ Stromal (CAF) Dkk-3 is pro-tumorigenic	Prognostic marker associated with poor outcome.	[17]
**Ovarian**	Significantly upregulated in ovarian cancer tissue Stromal Dkk-3 is pro-tumorigenic	Prognostic marker associated with poor outcome.	[17]
**Colon**	Significantly upregulated in colon cancer tissue Stromal Dkk-3 is pro-tumorigenic	Prognostic marker associated with poor outcome.	[17]
**Prostate**	➢ Increased Dkk-3 expression in prostate stroma attenuates TGF-beta signaling, supports formation of prostate epithelial cell acinar formation and reduces cancer cell proliferation in vitro. ➢ Promotion of tumor-associated antigen-specific CD8^+^ cytotoxic T lymphocytes (CTLs).	Prognostic marker associated with good outcome.	[9,13,18]
	Increased Dkk-3 expression in prostate stroma promotes stromal cell proliferation, promotes fibroblast to myofibroblast differentiation, and contributes to the angiogenic switch by suppressing vessel-stabilizing angiogenic factors like ANGPT1.	Prognostic marker associated with poor outcome.	[18]
**Pancreas**	Secretion of Dkk-3, primarily by pancreatic stellate cells, increases pancreatic ductal adenocarcinoma growth in vitro and in vivo, and induces chemo-resistance.	Prognostic marker associated with poor outcome.	[19]
**Medulloblastoma**	Non-MYC-driven medulloblastoma; DKK3 enhances new vascularization.	Therapeutic potential and progression biomarker.	[20]

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
