# Peer review of "Dickkopf-3: An Update on a Potential Regulator of the Tumor Microenvironment"

_cancers, 2022, doi:10.3390/cancers14235822_

Round 1
Reviewer 1 Report
This review is clear and comprehensive. Unlike a dozen of previous review papers of DKK3, the unique point is the reviewing tumor microenvironment. However, most of figures are not associated with tumor microenvironment except Fig, 1 and 4. I would like to recommend to omit Table 1 and Figure 2, and instead, to add microenvironment related table or figures.
In page 3: ‘This interaction was first observed in a yeast-2-hybrid screen [32] but its significance was unclear, given that Dkk-3 is a secreted protein.’
è DKK3 is secreted protein, however it obviously gets into the cytoplasm and downregulate beta-catenin [PMID: 35205672, Supplementary data of PMID: 19003969]. Please revise the sentence based on these studies.
In page 4: ‘At present, there are few examples of partners for the secreted form of Dkk-3.’
è Secreted DKK3 has been evaluated in the previous study (PMID: 35205672), showing tumor suppressor property of ovarian cancer. Please revise the sentence based on this study.
Figure 1: This figure seems like that DKK3 is involved in only three areas. However, as you know, many studies revealed the roles of DKK3 in cancer cell. So, please modify the figure or the legend to clarify it.
Table 1:
1) This summary is limited. The previous review papers are more informative than this table. In addition, this contents of Table 1 is not regarding of microenvironment. Is it really necessary in this review?
2) What does most common cancers in human mean? Based on incidence? In USA?
3) What does the role of DKK3 mean? The role in cancer cell? or in tissues? In clinical significance in patients?
Figure 2: What is the point of this figure? This is not new.
Figure 4: Regarding the binding of DKK3 with Kremen, there are three more reports, which I summarized below. Based on these, modification of figure could be needed.
· DKK3 does not interact with Kremen on the cell surface [PMID: 12050670; PMID: 12527209]. Instead, DKK3 colocalizes with Kremen-1 at the intracellular membranous compartment, such as the Golgi apparatus or the endoplasmic reticulum [PMID: 20370576].
Author Response
This review is clear and comprehensive. Unlike a dozen of previous review papers of DKK3, the unique point is the reviewing tumor microenvironment.
Comment #1: However, most of figures are not associated with tumor microenvironment except Fig, 1 and 4. I would like to recommend omitting Table 1 and Figure 2, and instead, to add microenvironment related table or figures.
Response: All comments raised by the reviewer have been taken in consideration. Table 1 has been modified as suggested to reflect tumor microenvironment more. Figure 1 has been also modified to clarify more data. Figure 2 has been removed as suggested.
Comment #2: On page 3: ‘This interaction was first observed in a yeast-2-hybrid screen [32] but its significance was unclear, given that Dkk-3 is a secreted protein.’
Response: The sentence was modified as suggested
Comment #3: DKK3 is secreted protein, however it obviously gets into the cytoplasm and downregulate beta-catenin [PMID: 35205672, Supplementary data of PMID: 19003969]. Please revise the sentence based on these studies.
Response: It has been modified and more data have been included.
Comment #4: In page 4: ‘At present, there are few examples of partners for the secreted form of Dkk-3.’
Response: More examples have been added now, while more investigation still needs to be included for future studies.
Comment # 5: Secreted DKK3 has been evaluated in the previous study (PMID: 35205672), showing tumor suppressor property of ovarian cancer. Please revise the sentence based on this study.
Response: It has been revised as suggested. Thank you
Comment #6: Figure 1: This figure seems like that DKK3 is involved in only three areas. However, as you know, many studies revealed the roles of DKK3 in cancer cells. So, please modify the figure or the legend to clarify it.
Response: The figure and the legend were modified.
Comment #7: Table 1:
1) This summary is limited. The previous review papers are more informative than this table. In addition, this contents of Table 1 is not regarding of microenvironment. Is it really necessary in this review?
2) What does most common cancers in human mean? Based on incidence? In USA?
3) What does the role of DKK3 mean? The role in cancer cell? or in tissues? In clinical significance in patients?
Response: The table was modified per the reviewer’s comment. It has been focused on the DKK-3 in the microenvironment.
Comment #8: Figure 2: What is the point of this figure? This is not new.
Response: It has been removed
Comment #9: Figure 4: Regarding the binding of DKK3 with Kremen, there are three more reports, which I summarized below. Based on these, modification of figure could be needed.
Response: The figure has been modified and more data have been included. It is Figure 3 now
Comment #10: DKK3 does not interact with Kremen on the cell surface [PMID: 12050670; PMID: 12527209]. Instead, DKK3 colocalizes with Kremen-1 at the intracellular membranous compartment, such as the Golgi apparatus or the endoplasmic reticulum [PMID: 20370576].
Response: It has been corrected. Thank you
Reviewer 2 Report
The authors present a review article in which they discuss the role of Dickkopf-3 (DKK3/REIC) as a potential regulator of the tumor microenvironment. By examining the literature, there are no other review articles examining this topic in such a comprehensive manner.
The abstract provides a concise overview of the article, and sets the main aim - to describe both pro- and antitumorigenic effects of endogenous Dickkopf-3. The wording “Dkk-3 is a divergent member of” should perhaps be replaced with “Dickkopf-3 (Dkk-3) is a protein in the Dickkopf family”, to avoid starting the abstract with an abbreviation, as well as to avoid the exact wording as in some other articles on DKK3 (Romero 2013., Hamzehzadeh 2018.).
The introduction explores the role of DKK3 in tumor suppression and promotion, and gives an overview of its role in the most common human cancers (Table 1) - I believe the dual, context-dependent nature of DKK3 (tumor suppression vs. promotion) would be better visualized if the table had two columns - 1) e.g. “tumor suppressing activity” and 2) e.g. “tumor promoting activity” (such as in ref. 18, 50 etc.).
The authors then discuss the specific mechanisms of tumor suppression. They state that “A variety of signaling pathways are affected by Dkk-3, accounting for both oncogenic and tumor-suppressive properties” and provide the reference 14 (Gondkar et al.), but it is unclear from the text (and the reference), which pathway other than Wnt/beta-catenin is the text referring to - the authors’ idea would perhaps be more clearly communicated if this section had a more funnel-shaped format.
They continue by discussing the interaction between DKK3 and various components of the tumor microenvironment (stromal cells, neoangiogenesis, immune cells and stem cells), and give numerous examples and references for each component. Although some examples do not pertain to tumor microenvironment per se (diabetic patients with erectile dysfunction, osteoarthritis etc.), the mechanisms described herein are clearly applicable in the setting of cancer.
They conclude by providing future perspectives in cancer therapy, focused on DKK3, and mentioning currently ongoing trials in this domain.
Minor notes:
- Page 1, Paragraph 1, Line 14: ER abbreviation should be preceded by endoplasmic reticulum
- Page 3, Paragraph 2, Line 7: β-TrCP - Beta-transducin repeats-containing proteins should be added before the abbreviation
- Page 4, Paragraph 4, Line 15: EMT - epithelial-mesenchymal transition abbreviation
- Page 7, Paragraph 1, Line 9: HUVEC - Human umbilical vein endothelial cells abbreviation
Author Response
The authors present a review article in which they discuss the role of Dickkopf-3 (DKK3/REIC) as a potential regulator of the tumor microenvironment. By examining the literature, there are no other review articles examining this topic in such a comprehensive manner.
Comment #1: The abstract provides a concise overview of the article, and sets the main aim - to describe both pro- and antitumorigenic effects of endogenous Dickkopf-3. The wording “Dkk-3 is a divergent member of” should perhaps be replaced with “Dickkopf-3 (Dkk-3) is a protein in the Dickkopf family”, to avoid starting the abstract with an abbreviation, as well as to avoid the exact wording as in some other articles on DKK3 (Romero 2013., Hamzehzadeh 2018.).
Response: It has been modified. Thank you
Comment #2: The introduction explores the role of DKK3 in tumor suppression and promotion, and gives an overview of its role in the most common human cancers (Table 1) - I believe the dual, context-dependent nature of DKK3 (tumor suppression vs. promotion) would be better visualized if the table had two columns - 1) e.g. “tumor suppressing activity” and 2) e.g. “tumor promoting activity” (such as in ref. 18, 50 etc.).
Response: The information required by the reviewer has been included in Table 1 and within the manuscript. Other articles such as Irina Giralt et al, 2014, which is included in our manuscript under citation [33], extensively addresses this point too.
Comment #3: The authors then discuss the specific mechanisms of tumor suppression. They state that “A variety of signaling pathways are affected by Dkk-3, accounting for both oncogenic and tumor-suppressive properties” and provide the reference 14 (Gondkar et al.), but it is unclear from the text (and the reference), which pathway other than Wnt/beta-catenin is the text referring to - the authors’ idea would perhaps be more clearly communicated if this section had a more funnel-shaped format.
Response: Thank you very much. We explained in brief how DKK-3 mediates its function either through Wnt signaling or Wnt independent pathway. This is followed by its tissue specific DKK-3/signaling activation. It has been modified now
Comment #4: They continue by discussing the interaction between DKK3 and various components of the tumor microenvironment (stromal cells, neoangiogenesis, immune cells and stem cells), and give numerous examples and references for each component. Although some examples do not pertain to tumor microenvironment per se (diabetic patients with erectile dysfunction, osteoarthritis etc.), the mechanisms described herein are clearly applicable in the setting of cancer.
Response: Thank you for your comment, examples not related to the tumor microenvironment have been removed.
Comment #5: They conclude by providing future perspectives in cancer therapy, focused on DKK3, and mentioning currently ongoing trials in this domain.
Response: Yes, Thank you
Comment #6: Minor notes:
Comment #5-1: Page 1, Paragraph 1, Line 14: ER abbreviation should be preceded by endoplasmic reticulum
Response: It has been included
Comment #5-2: Page 3, Paragraph 2, Line 7: β-TrCP - Beta-transducin repeats-containing proteins should be added before the abbreviation
Response: It has been included
Comment #5-3: Page 4, Paragraph 4, Line 15: EMT - epithelial-mesenchymal transition abbreviation
Response: It has been included
Comment #5-4: Page 7, Paragraph 1, Line 9: HUVEC - Human umbilical vein endothelial cells abbreviation
Response: It has been included
Reviewer 3 Report
The review article titled “ Dickkopf-3: an update on a potential regulator of the tumor mi-croenvironment” is talking about the cancr angiogenesis, immune cell response, and in stem cell differenctiation, and its epigenetic modeling of is therapeutic perspective. This review article is well written and cover major aspects of expressin and regulation of DKK3 in cance regulation.
This review article lacks the authors' opinion or input and a basic literature review. The pointwise comments are as follows.
1. What are the limitation of the DKK3 in cancer therapeutics?
2. How DKK3 is criticle in the organ fibrosis?
3. What are the other therapeutic aspects of the DKK3 in cancer progression?
4. Is there any human case study where DKK3 was tested for its modulation in the expression?
Author Response
The review article titled “ Dickkopf-3: an update on a potential regulator of the tumor mi-croenvironment” is talking about the cancr angiogenesis, immune cell response, and in stem cell differenctiation, and its epigenetic modeling of is therapeutic perspective. This review article is well written and cover major aspects of expressin and regulation of DKK3 in cance regulation.
Comment: This review article lacks the authors' opinion or input and a basic literature review. The pointwise comments are as follows.
Response: We have modified the review according to the reviewer’s comment. Thank you
Comment #1: What are the limitation of the DKK3 in cancer therapeutics?
Response: Limitations have been included now
Comment #2: How DKK3 is criticle in the organ fibrosis?
Response: It has been added
Comment #3: What are the other therapeutic aspects of the DKK3 in cancer progression?
Response: Another section describing therapeutic aspects of DKK3 has been included.
Comment #4: Is there any human case study where DKK3 was tested for its modulation in the expression?
Response: Yes, and it has been mentioned
Reviewer 4 Report
Authors provide a detailed summary of multiple studies linked to Dickkopf-3 in regulation of tumor microenvironment. Authors have provided a detailed description of role of Dickkopf-3 as a tumor suppressor as well as its pro tumorigenic roles. The review includes latest findings from the field and provides a comprehensive view of the role of Dickkopf-3 in cancer.
Comments regarding the submission.
1. The studies cited in this review have been systematically put forward and the outcome of all the studies have been well summarized.
2. Scientifically correct conclusions have been drawn when combining various independent studies to make a collective scientific statement.
3. The table included in the review provides an opportunity to a have quick and concise look at the role of Dickkopf-3 in various cancer types.
4. The review article has been neatly written with acronyms and abbreviations properly explained for greater accessibility.
5. The review is a novel summary of role of Dickkopf-3 in cancer stromal cell regulation, angiogenesis and cancer immune response
6. Overall, the review is a good read which provides a critical summary of the topic discussed.
Author Response
Authors provide a detailed summary of multiple studies linked to Dickkopf-3 in regulation of tumor microenvironment. Authors have provided a detailed description of role of Dickkopf-3 as a tumor suppressor as well as its pro tumorigenic roles. The review includes latest findings from the field and provides a comprehensive view of the role of Dickkopf-3 in cancer.
Comments regarding the submission.
Comment #1: The studies cited in this review have been systematically put forward and the outcome of all the studies have been well summarized.
Response: Thank you
Comment #2: Scientifically correct conclusions have been drawn when combining various independent studies to make a collective scientific statement.
Response: Thank you
Comment #3: The table included in the review provides an opportunity to a have quick and concise look at the role of Dickkopf-3 in various cancer types.
Response: Thank you, however it has been modified to address other reviewers’ comments
Comment #4: The review article has been neatly written with acronyms and abbreviations properly explained for greater accessibility.
Response: Thank you
Comment #5: The review is a novel summary of role of Dickkopf-3 in cancer stromal cell regulation, angiogenesis and cancer immune response.
Response: Thank you
Comment #6: Overall, the review is a good read which provides a critical summary of the topic discussed.
Response: Thank you and appreciate your kind feedback.